

# Spatial modeling could not differentiate early SARS-CoV-2 cases from the distribution of humans on the basis of climate in the United States

Robert Harbert[1,2], Seth W. Cunningham[2,3] and Michael Tessler[2,4]

[1] Biology, Stonehill College, Easton, MA, USA
[2] Sackler Institute for Comparative Genomics, American Museum of Natural History, New York, NY, USA
[3] Department of Biological Sciences, Fordham University, Bronx, NY, USA
[4] Department of Biology, St. Francis College, Brooklyn, NY, USA

Corresponding authors
Robert Harbert,
rharbert@stonehill.edu
Michael Tessler, mtessler@sfc.edu

## ABSTRACT

The SARS-CoV-2 coronavirus is wreaking havoc globally, yet, as a novel pathogen, knowledge of its biology is still emerging. Climate and seasonality influence the distributions of many diseases, and studies suggest at least some link between SARS-CoV-2 and weather. One such study, building species distribution models (SDMs), predicted SARS-CoV-2 risk may remain concentrated in the Northern Hemisphere, shifting northward in summer months. Others have highlighted issues with SARS-CoV-2 SDMs, notably: the primary niche of the virus is the host it infects, climate may be a weak distributional predictor, global prevalence data have issues, and the virus is not in population equilibrium. While these issues should be considered, we believe climate's relationship with SARS-CoV-2 is still worth exploring, as it may have some impact on the distribution of cases. To further examine if there is a link to climate, we build model projections with raw SARS-CoV-2 case data and population-scaled case data in the USA. The case data were from across March 2020, before large travel restrictions and public health policies were impacting cases across the country. We show that SDMs built from population-scaled case data cannot be distinguished from control models (built from raw human population data), while SDMs built on raw case data fail to predict the known distribution of cases in the U.S. from March. The population-scaled analyses indicate that climate did not play a central role in early U.S. viral distribution and that human population density was likely the primary driver. We do find slightly more population-scaled viral cases in cooler areas. Ultimately, the temporal and geographic constraints on this study mean that we cannot rule out climate as a partial driver of the SARS-CoV-2 distribution. Climate's role on SARS-CoV-2 should continue to be cautiously examined, but at this time we should assume that SARS-CoV-2 will continue to spread anywhere in the U.S. where governmental policy does not prevent spread.

## INTRODUCTION

### SARS-CoV-2 and climate overview

In March 2020, the USA had the most reported cases of COVID-19, the disease caused by the coronavirus SARS-CoV-2 (*CDC NCIRD, 2020*). While we are now coming to understand the global and US-level distributions of this virus, we are only starting to learn how abiotic variables affected its geographic distribution, particularly before policy seemed to become the principal driver of the virus. With over two months of SARS-CoV-2 records across U.S. counties by the end of March 2020, the role of abiotic variables had become easier to examine, especially as multiple sources began compiling data and making it public (*The New York Times, 2020*; *Dong, Du & Gardner, 2020*; *The COVID Tracking Project, 2020*).

Organisms are distributed within their environment based on both direct and indirect interactions with biotic and abiotic variables (*Elith & Leathwick, 2009*)—SARS-CoV-2 is no exception. As this virus is primarily distributed by human hosts, it is constrained by human distributions and interactions (i.e., biotic variables). There are numerous documented cases on local and global scales of individual human sources for new outbreaks (*Holshue et al., 2020*; *KCDC, 2020*). Still, many other viruses, including other coronaviruses, display marked seasonality and are affected by local climatic conditions (*Price, Graham & Ramalingam, 2019*; *Fisman, 2012*; *Lofgren et al., 2007*; *Gaunt et al., 2010*). This has prompted researchers to begin looking globally at how weather and climate (i.e., a subset of abiotic variables) may relate to the presence and abundance of the virus.

Modeling the SARS-CoV-2 viral distribution from the early part of the outbreak in relation to climate and other abiotic variables could help refine our understanding of its spread, hopefully adding to the knowledge gleaned from studies on human transmission dynamics (*Kucharski et al., 2020*; *Chinazzi et al., 2020*), which appears to explain much of the viral distribution. Climate can also be compared directly to biotic variables, such as human population density, to disentangle their relative importances. More dense human populations can have elevated communicable disease spread, as there is more person-to-person contact (*Fang et al., 2013*)—the principal way SARS-CoV-2 is known to spread (*Rothan & Byrareddy, 2020*). Still, identification of abiotic variables with suitably large effect size on the distribution and spread of SARS-CoV-2 might help prevent viral spread by identifying higher-risk regions and projecting seasonal variation in the risk of transmission. However, it is important to make clear that the models presented here, and in other recent studies (*Araújo & Naimi, 2020*; *Bariotakis et al., 2020*), are influenced by the variability of input parameters, the variables explored, and the restrictions inherent in modeling any complex system (*Elith et al., 2011*). We present this study to serve as a baseline in our understanding, given the available data, rather than as a definitive model.

### SARS-CoV-2 and climate research

Early studies suggest that transmission of SARS-CoV-2 may have, at a minimum, a loose association with climatic features. There have been numerous reports showing a correlation of case incidence with cool temperatures and low humidity (*Wang et al., 2020*;

*Ficetola & Rubolini, 2020*; *Bannister-Tyrrell et al., 2020*; *Araújo & Naimi, 2020*; *Chen et al., 2020*; *Alvarez-Ramirez & Meraz, 2020*; *Notari, 2020*; *Paez et al., 2020*). Furthermore, studies that controlled for case growth rate or demographic factors found that weather was still a significant factor in the success of SARS-CoV-2 outbreaks (*Bukhari & Jameel, 2020*; *Chen et al., 2020*; *Sajadi et al., 2020*). Unfortunately, none of these studies provide a mechanism for how abiotic variables like temperature might limit or promote the person-to-person transmission of the virus. Still, lipid-enveloped viruses (including coronaviruses) may be more stable outside of the host in lower humidity and cooler temperatures, conditions that are common in temperate areas in late winter and early spring (*Price, Graham & Ramalingam, 2019*). Recent studies have also indicated that SARS-CoV-2 transmission rate may be negatively correlated with temperature and humidity (*Qi et al., 2020*; *Wu et al., 2020*; *Ahmadi et al., 2020*; *Rosario et al., 2020*; *Sagripanti & Lytle, 2020*). However, the mechanism causing these correlations could be indirect, as humans behave differently in different seasons (*Bedford et al., 2015*; *Wesolowski et al., 2017*). Recent epidemiological models have also highlighted that weather may not play a large role when most people lack immunity during the pandemic phase, but indicate that weather has the potential to play a larger role once the endemic phase is reached (*Baker et al., 2020*).

Biologists often employ species distribution models (SDMs; alternatively called ecological niche models) to predict geographic ranges of species. SDMs employ environmental data (typically climate) to predict if geographic space is suitable for a given species or population (*Peterson, 2001*; *Peterson et al., 2011*). These models have proven useful in a wide variety of applications, such as invasion biology, climate change, zoonotic diseases, and speciation (*Elith & Leathwick, 2009*; *Guisan & Thuiller, 2005*). SDMs built to predict the spread of viral pathogens often do so by modeling the potential distribution of known vectors or alternate hosts (*Miller et al., 2012*; *Larson et al., 2010*; *de Almeida et al., 2019*; *Richman et al., 2018*), but some base the models directly on pathogen occurrence data (*Machado-Machado, 2012*; *Belkhiria, Alkhamis & Martínez-López, 2016*; *Pigott et al., 2014a*; *Messina et al., 2015*; *Bhatt et al., 2013*). Efforts to model global disease distribution within an ecological framework frequently draw on known or hypothesized environmental variables correlated with disease occurrence to predict suitability for transmission. Output from these models can be extrapolated into areas where a disease has not yet been reported, but where suitable environmental variables correlated with its occurrence exist (*Elith & Leathwick, 2009*).

Researchers early in the pandemic created SDMs of SARS-CoV-2 using climatic variables (*Araújo & Naimi, 2020*; *Bariotakis et al., 2020*). These studies suggested that the virus is strongly constrained by global climate patterns (*Araújo & Naimi, 2020*; *Bariotakis et al., 2020*). Preliminary models also suggested that the virus would continue to be concentrated in the Northern Hemisphere, shifting northwards throughout the summer and then back towards its spring distribution in the fall and winter (*Araújo & Naimi, 2020*).

However, a research group put forth a strong rebuttal (*Chipperfield et al., 2020*) to some of this modeling work (*Araújo & Naimi, 2020*). Their main criticism asserts that basic assumptions underlying SDMs are violated when modeling SARS-CoV-2, due to the mode of transmission, current population disequilibrium, and failure to incorporate

epidemiological data (*Chipperfield et al., 2020*). Furthermore, they highlight issues pertaining to input data for global records, such as: missing data, omitted data, and single localities representing many thousands of reported cases (hospital coordinates or political centroids) across an entire country (*Araújo & Naimi, 2020*). This is exacerbated as many countries are reporting records based on varied criteria, such as multiple types of molecular tests and tests using computed tomography scans (*Lippi, Simundic & Plebani, 2020*). The rebuttal additionally suggested that model evaluation and justification were insufficient. An updated draft of the original paper (*Araújo & Naimi, 2020*) addressed some of these issues, and produced reasonable arguments to some others. While we agree with many arguments from the rebuttal, we believe SDMs have the potential to be useful for modeling in a number of instances, if done carefully, and have been effective when used for other diseases (*Pigott et al., 2014b*, *2015*; *Carlson, Dougherty & Getz, 2016*) and other instances when data are limited or incomplete (*Galante et al., 2018*; *Barbet-Massin et al., 2018*; *Katz & Zellmer, 2018*; *Pearson et al., 2006*; *Fois et al., 2018*; *Hernandez et al., 2006*; *Kiedrzyński et al., 2017*).

Still, the climate-based SDMs for SARS-CoV-2 presented recently may, to at least some extent, reflect the climatic preferences of their host. Careful consideration of host availability (human population density) and pathogen ecologies (abiotic variables related to transmission) may be necessary to frame analyses modeling global distributions (*Johnson, Escobar & Zambrana-Torrelio, 2019*), and may help to better ensure that projected distributions are not simply the result of environmental variables related to human population density.

## SARS-CoV-2 in the US

COVID-19 was first detected in the U.S. on 20 January 2020 (*Holshue et al., 2020*), has since been detected in all 50 states, and continues to spread rapidly (*Chinazzi et al., 2020*). Quality county-level data are also publically available (*The New York Times, 2020*; *The COVID Tracking Project, 2020*; *Dong, Du & Gardner, 2020*). As of 22 July 2020, SARS-CoV-2 has caused just under 145,000 deaths in the U.S. (*Murray & IHME COVID-19 team, 2020*). However, like much of the world, the case distribution is rapidly changing, and the case numbers continue to rise and fall. Public policy decisions appear to be the main mitigating factor against this coronavirus (*Leonhardt, 2020*).

Here we develop a suite of data visualizations, SDMs and comparative niche overlap analyses using both climate and human population data to determine whether the effect of climate can be appropriately disentangled from other drivers of SARS-CoV-2 transmission, given early records for the virus in the U.S. We feel that examining a relatively early time-point of the pandemic in the U.S. is useful, as it in part (probably largely) predates the impact of major shifts in public health policy for the U.S.

## MATERIALS AND METHODS

### Code and results deposition

The code and results developed in this study are deposited under a CC-BY-NC 4.0 License on GitHub (https://github.com/rsh249/cv19_enm/releases/tag/v0.0.5). All analysis code

was written using R 3.6.2 (*R Core Team, 2019*). Plots for Figs. 1–3, Figs. S1 and S2 were produced using the "ggplot2" package (*Wickham, 2009*).

## Data acquisition

SARS-CoV-2 case data for U.S. counties (one record per county, with a total sample size of 1,883 counties that had available reports) were collected from the New York Times database (*The New York Times, 2020*) on 31 March 2020, for the 30 March 2020 data release. County-level data on human population densities were acquired from the 2010 United States Census through the R "tidycensus" package (*Walker, Eberwein & Herman, 2020*). Georeferencing to county centroids was performed by referencing county and state names in the GeoNames database (https://www.geonames.org/). While exact virus case geolocations or even town-level data would provide finer scale resolution to our analyses, these data are likely the best curated dataset for the U.S. for this time period.

## Climate data

Interpolated climate data averaged from 1970 to 2000 for the month of March were accessed through the WorldClim v2.1 database (*Fick & Hijmans, 2017*). The seven climate parameters examined for the month of March were average monthly temperature, average monthly minimum temperature, average monthly maximum temperature, average monthly precipitation, average daily solar radiation, wind speed and water vapor pressure (a measure related to humidity). These climate parameters are consistent with possible correlates of SARS-CoV-2 transmission in several recent studies (*Qi et al., 2020*; *Sagripanti & Lytle, 2020*; *Ahmadi et al., 2020*; *Rosario et al., 2020*; *Wu et al., 2020*). Climate data were extracted from these layers for each georeferenced county record using the average from a 5 km buffer around each county centroid coordinate. These local averages should represent the county generally as climate is correlated over short distances, but these averages may not be the best representation in counties that have a large area or highly clustered population centers. While county centroids may not be ideal in all circumstances, our view is that county centroids plus this buffer will average the climate relative to many cases in the area. More precise case georeferencing is not possible without high levels of individual case and movement tracking to identify the source of infection.

## Climate distribution visualization

Total cumulative positive case data for SARS-CoV-2 in each reporting county on 30 March 2020, were used to extract data for the seven abiotic variables listed above. Probability density distributions for SARS-CoV-2 were produced to characterize the likelihood of case occurrence given the available range of climate values. These distributions are calculated from case occurrences and corresponding climate data using a Gaussian Kernel Density Estimator and standard bandwidth estimation. These distributions can be interpreted similarly to histograms, but are normalized to be comparable between different sample sizes. Raw case data (i.e., total cases per county) were scaled to reflect virus cases in each county unit by dividing total cases by the county population from the 2010 U.S. Census. These population-scaled viral cases (cases/population) were used as probability density

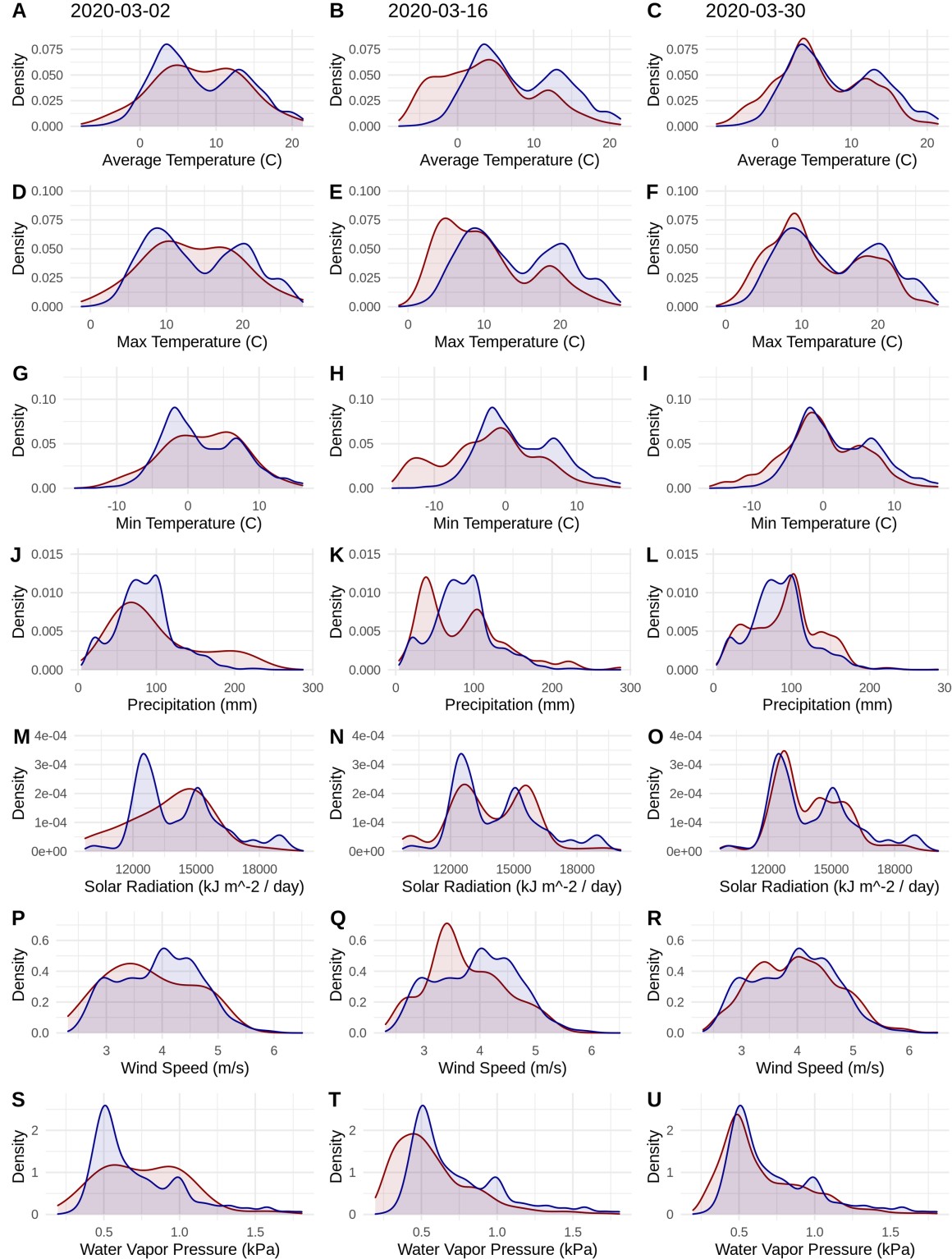

**Figure 1** Probability densities of SARS-CoV-2 coronavirus cases (using population scaled data; curves in red) compared to the probability densities of human populations (curves in blue) in each US county for each of seven climate variables (A–C) Average Temperature (C); (D–F) Max Temperature (C); (G–I) Minimum Temperature (C); (J–L) Precipitation (mm); (M–O) Solar Radiation (kJ m$^{-2}$/day); (P–R) Wind Speed (m/s); and (S–U) Water Vapor Pressure (kPa) at three time periods (March 3, 2020; March 16, 2020; March 30, 2020). Probability density curves are standardized to an area of one.

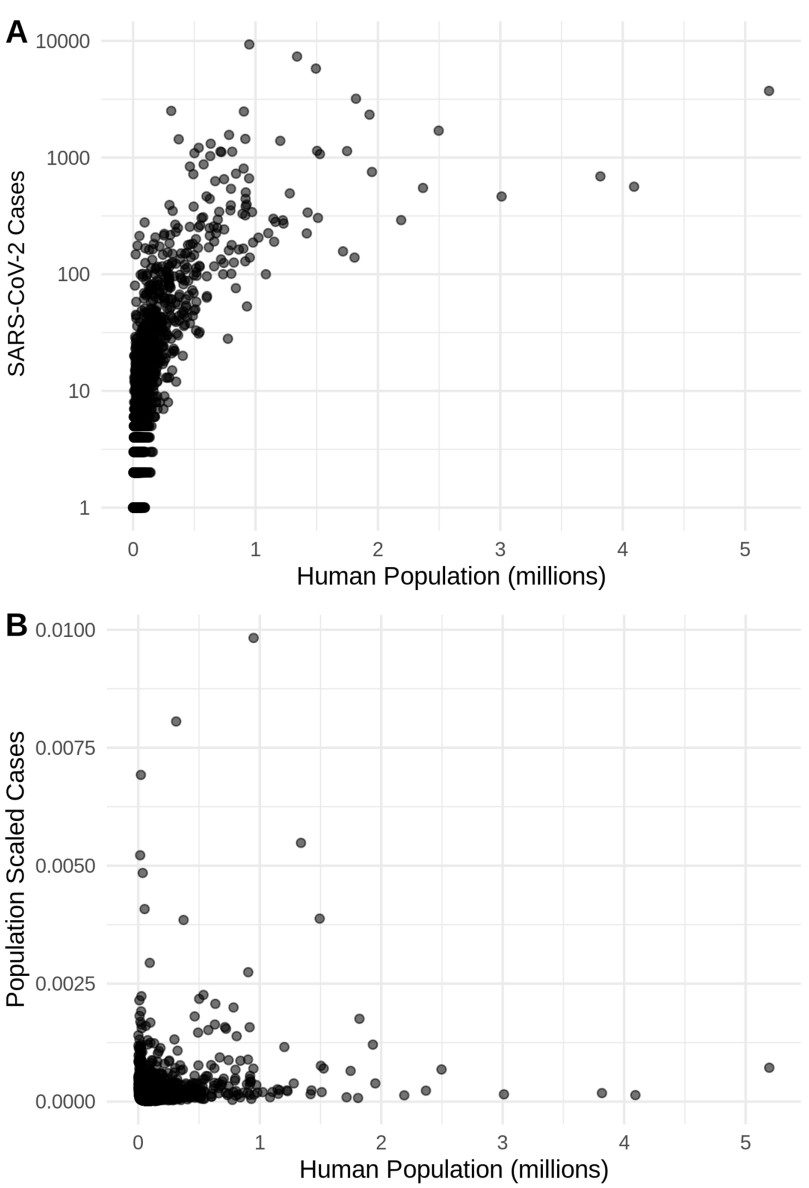

**Figure 2 The relationship in the US between human population size and SARS-CoV-2 coronavirus cases, using (A) total viral cases and (B) population scaled viral cases.** New York City, an outlying point, has been excluded for clearer visualization.

weightings, and resulting curves were standardized to an area of one. Probability densities were also calculated with raw case count data and county-level population as weightings. Probability density estimation and visualization was done with the R "ggplot2" package (*Wickham, 2009*). To test whether the SARS-CoV-2 cases and human population data were correlated, we applied Spearman tests "cor.test(method = "spearman")".

## Species distribution models

Occurrence data for raw (i.e., total viral cases) SDMs were generated by expanding each county climate record by multiplying that occurrence by the total SARS-CoV-2 case count
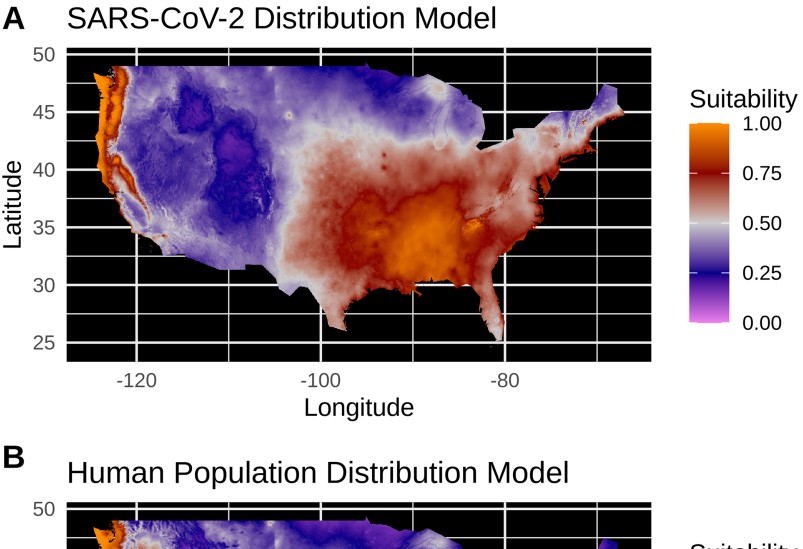

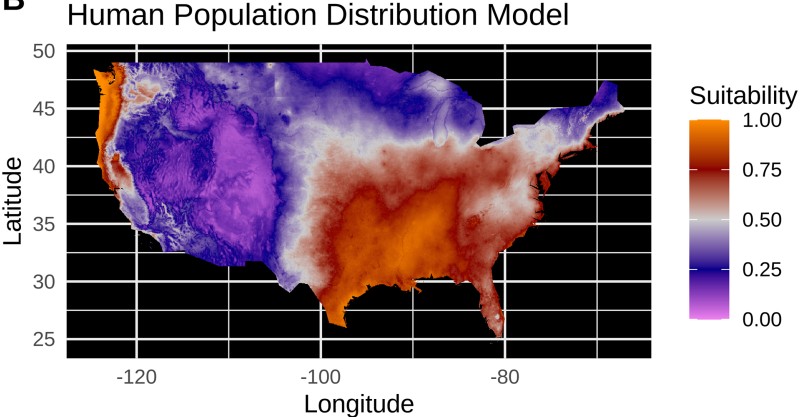

**Figure 3** (A) Species distribution model of the SARS-CoV-2 coronavirus (using population scaled data) for 30 March 2020. (B) Human population distribution model for the US from 2010.

for each county. Occurrence data for population-scaled SDMs were generated the same way as for the raw SDMs except that county climate records were multiplied by the total case count divided by the county population. Then population-scaled values were multiplied by an expansion factor of 100,000 so that all counties with at least one case were represented. More explicitly: raw data = total viral cases; population-scaled data = total viral cases/human population × 100,000 (the expansion factor for our data). Viral host availability (i.e., total humans per county) was modeled to serve as a null distribution to be used as a control comparison for viral SDMs.

Maxent, an algorithm for presence-only distribution models (*Elith et al., 2011*), parameters relating to model complexity were tested by building a suite of SDMs for SARS-CoV-2 distribution data with occurrences generated from raw reported virus case values and population-scaled case values. Maxent model testing and cross validation was performed using the ENMeval package (*Muscarella et al., 2014*) considering linear and quadratic feature classes (constraints on model fit) and regularization multipliers (penalties on complexity) of 0.5, 1, 1.5, 2, 2.5 and 3. Optimal model parameters were chosen by maximizing the average test AUC calculated with cross validation using the spatial "block" partitioning method, and minimizing AICc in the case of ties. All models

were built using the Maxent algorithm implemented in the "maxnet" R package (*Phillips, 2017*), including those tested within ENMeval. Operational models using the optimal model parameters were then built for SARS-CoV-2 using all population-scaled data, raw virus data, and for the human population using the county population data (*Phillips, 2017*). Of the seven WorldClim variables, the three temperature variables are not independent. Accordingly, we only used maximum temperature, as higher temperatures have been hypothesized to lower the viral distribution. Models built using all temperature variables made similar models, but are not further reported on. Reducing correlated variables is generally preferable to avoid extrapolation issues that arise from local covariance between variables (*Araújo et al., 2019*).

Given recent critiques, it has become clear that when using SDMs in relation to SARS-CoV-2, it is worth documenting how close a study may come to the gold standards set for this type of modeling (*Araújo et al., 2019*). The mode rank for our SDMs is bronze (Table 1), largely as data from an emerging disease are inherently imperfect. Our SDMs rely on incomplete (limited or underreported data) observational data that reports only positive tests for the continental United States. The SDMs also only model over climate averages for the month of March (averaged from 1970 to 2000), which may result in a truncated climate envelope. However, despite imperfect data the SDMs reported here fulfill the silver or gold standard methodology for model complexity, treatment of bias and model evaluation (Table 1).

Niche overlap and similarity tests were conducted with the "ecospat" library (*Di Cola et al., 2017*; *Broennimann et al., 2012*; *Warren, Glor & Turelli, 2008*) to compare the climatic niche of SARS-CoV-2 and humans. The niche overlap test considers whether there is a greater than expected overlap between the climate space occupied by two population models (human total vs. SARS-CoV-2 positive) than would be expected given a null distribution sampled from both populations. The niche similarity test considers the degree of similarity between the density of occurrence between two population models relative to the same null distribution (*Warren, Glor & Turelli, 2008*). A significant overlap between SARS-CoV-2 and total human population models indicates that the virus' distribution is not necessarily constrained by climate (i.e., that the full range of human occupied climate is accessible to the virus).

# RESULTS

## Climate distribution visualizations

The human population climate curves are visually similar compared to the population-scaled SARS-CoV-2 climate curves for all seven abiotic variables for March 30 (Fig. 1). There do appear to be modest differences in the population-scaled viral curves towards cooler temperatures and lower water vapor pressure compared to the human population curves. The raw density of SARS-CoV-2 cases (Fig. S1) is visually different from the population-scaled data (Fig. 1). Population-scaled SARS-CoV-2 data appear to become better fitted to the human population data over time (i.e., these curves better match on March 30 than they did on March 2 or 16, 2020; Fig. 1), whereas the raw SARS-CoV-2 case data become more and more fixed on a narrow range of values for all

**Table 1 Evaluation of our species distribution modeling practices against the best practices that have been proposed for this field (*Araújo et al., 2019*).**

| Guideline | Standard | Justification |
|---|---|---|
| Response variables | (A) Sampling: bronze | Best data available; municipalities, local governments, and states choose who to test. Positive tests only reported |
| | (B) Identification: gold | Assuming best practices in testing and reporting |
| | (C) Spatial accuracy: bronze | County assignments provide a rough georeference for each record, but do not precisely describe where transmission of the virus occurred. Spatial accuracy unknown. Occurrences limited to identifiable county level localities |
| | (D) Environmental extent: deficient | Limiting the study area to the continental U.S. is unlikely to adequately test environmental boundaries |
| | (E) Geographic extent: bronze | Study area to include current range in the U.S. |
| Predictor variables | (A) Selection of candidates: bronze/deficient | Unclear and not well documented correlations between SARS-CoV-2 transmission and climate. At best, distal variables with weak, indirect control on the distribution |
| | (B) Spatial and temporal resolution: deficient | Variables sampled from a 2.5 arcminute grid for all cells within 5km of each occurrence point. Mean value used for modeling. Monthly climate averages as predictors for end of March occurrence data |
| | (C) Uncertainty: bronze | Temporal and spatial uncertainty in occurrence data has unquantified potential effects on the model output. |
| | (A) Model Complexity: silver | ENMeval for model testing and selection (maximize testing AUC and minimize AICc in the case of ties) using internal cross validation through the block resampling method |
| | (B) Treatment of response bias: silver | Internal cross validation to evaluate bias effects in different models |
| | (C) Treatment of collinearity: bronze | "Approximate methods are applied" — Predictor variables hand selected from monthly climate data available to avoid collinearity (i.e., used only Tmax and not Tavg or Tmin) |
| | (D) Uncertainty: bronze | Multiple Maxent model parameters tested, but only the optimal model presented |
| Model evaluation | (A) Evaluation of model assumptions: gold/silver | Select robust models from all tested models with ENMeval |
| | (B) Evaluation of model outputs: silver | Evaluated against multiple, non-independent, geographically structured sub-samples |
| | (C) Measures of model performance: silver | Suite of model performance metrics performed via ENMeval |
| Summary | Mode of the scores: bronze | Model building and testing is generally robust, but data and geographic scope are incomplete at this time |

climate variables (Fig. S1). The latter (fixed and narrow) range presumably has to do with a few virus hotspots.

There is a highly significant positive relationship (Spearman's $r = 0.75$, $p < 2\text{E}{-}12$); between the number of humans in a county and the number of viral cases; however, the population-scaled viral cases do not have a strong relationship (Spearman's $r = 0.05$, $p = 0.03$) with human populations (Fig. 2). Although the latter $p$-value is significant, this is often not all that meaningful for large datasets, whereas the weak $r$ value better illustrates the relationship for the population-scaled data.

## Species distribution models

Model testing for SARS-CoV-2 population-scaled case model identified a Maxent model with linear and quadratic feature classes and a regularization multiplier of 0.5 as the best model, with a reasonably high model fit and transferability under block resampling

(Avg. Test AUC = 0.82; Test AUC variance = 0.0002). Population-scaled case and human population models built using these optimal model parameters appear to be highly similar, with areas of high suitability predicted for much of the West Coast and most of the eastern half of the U.S. (Fig. 3). There are some differences; for instance, the population-scaled model output for SARS-CoV-2 (Fig. 3A) suggests Florida is less suitable than it is in the map for human population (Fig. 3B).

Model testing for the SARS-CoV-2 raw case dataset identified a Maxent model with linear and quadratic feature classes and a regularization multiplier of 1.5 as the optimal model with a high model fit (Avg. Test AUC = 0.88), but with a slightly lower model transferability evidenced by higher AUC variance (Test AUC variance = 0.007) than the population-scaled model. The SDMs of the raw virus cases (Fig. S2) fail to reconstruct the known distribution of viral cases in much of the U.S. with strong bias towards the Pacific Northwest and Northeastern United States. The SDMs for humans (Fig. 3; Fig. S2) do not exactly match current human distribution for the U.S. However, this was expected, because the goal of the model is to match climates that are correlated with the places where most people live (i.e., it is not simply a map of current human populations). We did this as our ultimate goal was comparing a viral model to a human model.

The niche overlap and similarity tests both find that the models of SARS-CoV-2 (population-scaled viral case data) and human population have significantly higher overlap ($p < 0.01$) than would be expected by chance (Fig. 4), confirming the visual similarities between these maps. The raw data showed less overlap: the similarity test was significant, but an order of magnitude less so than for the population-scaled data ($p = 0.02$), while the overlap test was insignificant ($p = 0.20$), as seen in Fig. S3.

## DISCUSSION

### Main findings

Our results suggest that population-scaled SARS-CoV-2 coronavirus cases from March 2020 were highly linked with the human populations in the U.S. and that any influence of climate is hard to disentangle for SARS-CoV-2 cases for the U.S. during this time period (Figs. 1–4). Furthermore, this link was strengthening over the duration of March (Fig. 1). This indicates that caution should be used when dealing with climate modeling of SARS-CoV-2, at least on a national scale. Our results, while broadly similar to what was found in the U.S. for global-scale models (*Araújo & Naimi, 2020*; *Bariotakis et al., 2020*), indicate that the pattern we observed is simply reflecting human population density. Based on this, at least human population density should be compared or incorporated in any modeling exercises for SARS-CoV-2. Accordingly, the current global results from other studies (*Araújo & Naimi, 2020*; *Bariotakis et al., 2020*) should not be taken at face value without a critical comparison to human distributions. Furthermore, this is in line with current knowledge that public policy is the main driver for coronavirus spread in the U.S. (*Leonhardt, 2020*). It is also in line with modeling efforts that have combined epidemiological variables with climate (*Baker et al., 2020*).

It is also noteworthy that our models using the raw viral case data (Fig. S2), as opposed to population-scaled data (Fig. 3), performed poorly: the raw SDMs did not highlight

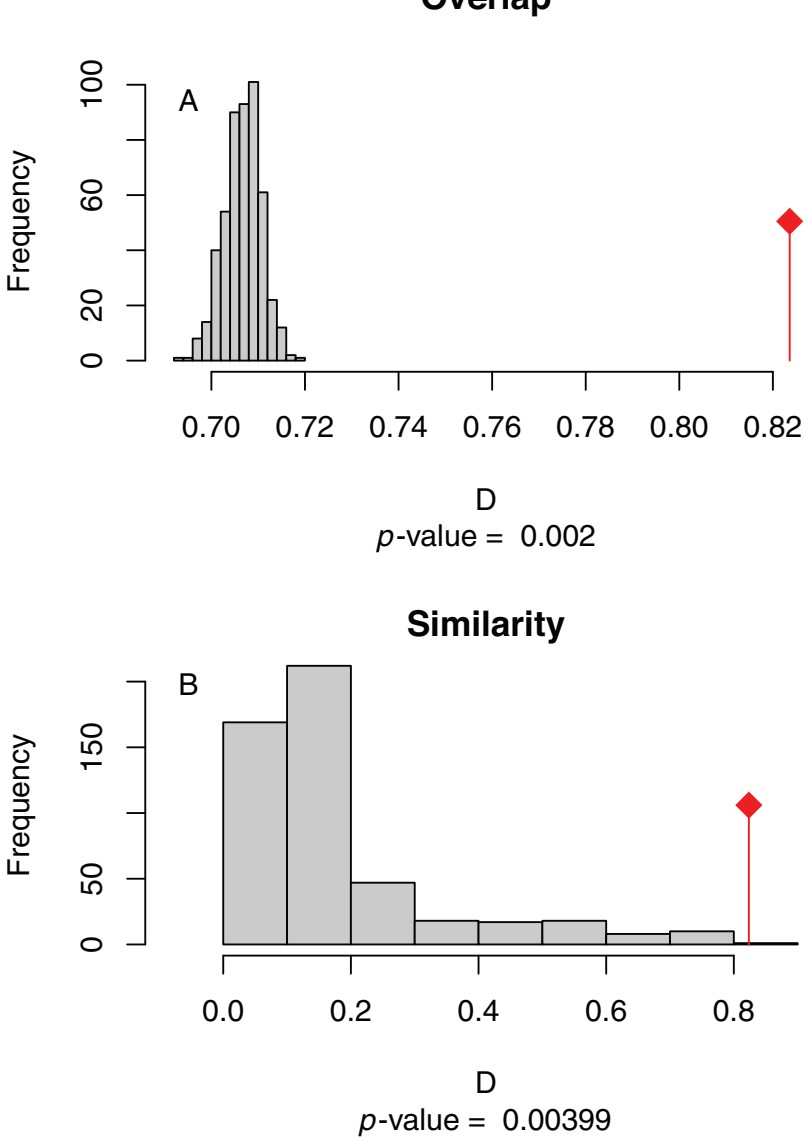

**Figure 4** (A) Niche overlap and (B) similarity tests for Maxent species distribution models built with population scaled SARS-CoV-2 coronavirus data compared to one built with human population density as occurrence data; actual model overlap indicated by a red marker in both plots. Significant *p*-values correspond to greater niche overlap or similarity than expected by random models.

most of the known March range of SARS-CoV-2 in the U.S. Other distribution modeling studies have only used raw values as inputs (*Araújo & Naimi, 2020*; *Bariotakis et al., 2020*), which our model shows to be at risk of overfitting and low model transferability to novel regions. Similarly, our analysis of raw SARS-CoV-2 case data shows that over time the probability density curves focus on a narrower range of climate as more data comes from the areas most affected by the outbreak (Fig. S1). In contrast, adjusting for population size with the population scaling results in SARS-CoV-2 climate curves more closely approaches that of the human population curves over time (Fig. 1). Furthermore,

county-level human population numbers only appear to be strongly correlated with raw cases, rather than population-scaled cases (Fig. 2). Together these observations suggest that SARS-CoV-2 was highly successful in a few areas in March (e.g., New York City), but when scaled for the available human population in each county we can infer a much broader geographic and climatic range was available.

Again, our data provide a few indications that climate may have had some relevance for SARS-CoV-2 distributions in March. Viral probability densities were slightly higher in cooler locales (Fig. 1; Fig. S1). The population-scaled data were particularly compelling, even if the climate difference is smaller, as they are less driven by the current SARS-CoV-2 hotspots. Furthermore, our population-scaled viral SDMs, while statistically indistinguishable from our human SDMs, had lower suitability for the virus in Florida and the southwestern border (Fig. 3). Whereas in reality, Florida had >5,000 cases in March.

## Avoiding a presumably faulty prediction

Given that our SDMs for humans and the virus had significant niche overlap and similarity (Fig. 4), we do not believe that a future projection of our SARS-CoV-2 SDMs in the U.S. using only climate data would be trustworthy at this point in time, and therefore we do not present one. A model based on March data would likely have suggested the SARS-CoV-2 prevalence would have shifted northward during the summer, which would be congruent with another early global modeling study (*Araújo & Naimi, 2020*). However, given that humans do not move northward in large numbers in the U.S. during the summer, our predictions would be based on faulty presumptions about host resources (i.e., that there are sufficient humans in the north to harbor the bulk of cases). Most importantly, we now know that the virus has spread across much of the southern U.S. during the summer. One could imagine this counterintuitive spread was, in addition to inadequate public health policy, due in part to people in the southern states staying indoors more in the summer, replicating the behavior found in the northern states during the winter. Still, it is possible that a future model projection would turn out to be correct using updated datasets, and there could be at least some residual predictive power that is not fully encompassed by human population patterns.

## Modeling counterarguments

Here are some of the main arguments that have been put forward *against* SARS-CoV-2 SDMs, followed by our opinion on the subject. (1) Issue: the virus is spreading and the population was not in equilibrium with climate or any other putative niche dimension in March, let alone today. Our thoughts: it was not in equilibrium and that is imperfect; however, SDMs have been useful for a number of non-equilibrium systems, like species invasions during their spreading phase (*Václavík & Meentemeyer, 2009*), and waiting for equilibrium will mean predictions are no longer as useful to conduct (i.e., would only be helpful for a next outbreak of this virus). (2) Issue: the virus may have been spreading heavily or underreported in the Global South as of March. Our thoughts: surely this was partially true, but this may not completely explain observable climatic correlations. Even if we are working with incomplete data, it is hard to know if adding those data will

significantly change conclusions based on preliminary models (i.e., will the SARS-CoV-2 model still be indistinguishable from a human population model). Still, to mitigate this problem we focused on the more consistent US-level data. (3) Issue: papers applying SDMs must strive for best practices to avoid common errors. Our thoughts: yes, we agree that best practices are indeed worth pursuing, even if not everyone agrees on what best practices may be. We have done our best to achieve the somewhat aspirational standards that have been put forth (*Araújo et al., 2019*). We explicitly document this and summarize shortcomings that are inherent with a newly spreading system (Table 1). (4) Issue: caution should be taken in claims and dissemination of research that could impact public health policy. Our thoughts: yes, caution is advisable in terms of studies on dire subjects like SARS-CoV-2; however, cross pollination between disciplines has been important for many breakthroughs and advances in science.

It has now been argued that SDMs of SARS-CoV-2 can help to predict where the virus may generally be found now and in the future (*Araújo & Naimi, 2020*; *Bariotakis et al., 2020*), as well as why it might be a fool's errand to conduct these analyses in the first place (*Chipperfield et al., 2020*). Specifically, caution towards distribution modeling methodology for SARS-CoV-2 in recent review literature highlights the likely limited effect climate has on a pathogen spread via direct transmission, and thus concludes that this tool is inappropriate in this situation (*Carlson et al., 2020a*, *2020b*). Important confounding variables such as human interactions, public policy and microclimate may mask any effects climate plays on this virus (*Carlson et al., 2020b*; *Chipperfield et al., 2020*). Due to this, epidemiological studies are more suited to understand patterns of transmission (*Carlson et al., 2020a*, *2020b*; *Chipperfield et al., 2020*). Importantly, given these problems with SDMs of SARS-CoV-2, there is concern that these studies may negatively affect public policy (*Chipperfield et al., 2020*; *Carlson et al., 2020a*, *2020b*). Others continue to highlight the possible connections between climate and SARS-CoV-2 and advocate for continued research in this area (*Araújo, Mestre & Naimi, 2020*).

## Caveats and unknowns

There are individuals who regularly go undetected for SARS-CoV-2 (e.g., those who lack symptoms or have mild symptoms)—currently estimated at up to 25% (*Mandavilli, 2020*). This is of course an issue for all studies of SARS-CoV-2, but was likely especially true for March when testing was less widely available in the U.S. Obviously it is not ideal for modeling and forecasting; however, it is inherent in any study. In fact, we have far more data available for SARS-CoV-2 than we will ever have for the vast majority of viruses and biodiversity generally.

Our results are based on data for a single, large country. While conducting an analysis on global data would have been ideal, we avoided this because this type of global analysis is known to have certain issues and inconsistencies (*Chipperfield et al., 2020*). While it may somewhat hobble our ability to forecast viral distributions at other time points, as there are surely non-analogous weather systems to come (i.e., no place in the U.S. in March was as hot as Death Valley, California is during full summer heat), we feel it was a worthy trade-off. Policy, social factors and a variety of other variables were not included in

this study, despite their obvious, known, or potential importance (*Maier & Brockmann, 2020*; *Leung et al., 2020*; *Miller et al., 2020*); other variables such as wearing masks are also surely important for global patterns (*Maier & Brockmann, 2020*; *Leung et al., 2020*). Unfortunately, data on these variables are often lacking in public databases. Our goals were to focus on a rather macro scale, which may be harder to do with policy data that can vary substantially from county to county, state to state, and, for a global study, country to country. While policies in the U.S. probably had not been implemented for long enough to tell their impact in March data, these changes (e.g., self-isolation and mask wearing) clearly and strongly influenced the trajectory of the virus after March.

### Data accessibility and mobilization

We believe the New York Times (*The New York Times, 2020*) providing robust county-level data in an accessible repository with an open license for the U.S. sets an excellent standard that should be repeated by governments, academics and other media organizations for other parts of the world so that this type of study may be better repeated in any country or globally. In a similar vein, we have made our analytical and modeling pipeline (along with figures) available (https://github.com/rsh249/cv19_enm/releases/tag/v0.0.5). We believe it is imperative for all pipelines and scripts to be made available for any SARS-CoV-2 research to ensure that models can be improved upon and any errors can be more quickly uncovered and resolved.

### Future directions

There are many avenues to pursue regarding SARS-CoV-2 modeling and predictions. We are excited to see researchers from a variety of fields extending their toolkits towards understanding this virus. We hope that ecological studies like this and others can play a role without overcomplicating the research efforts put forth by epidemiologists. Still, studies should familiarize themselves with current critiques of SDMs for SARS-CoV-2 modeling and be cautious of their inputs and conclusions.

With improving data, we feel that future studies should better be able to examine the system globally while considering human populations and public policy efforts at curbing the virus. We also believe that it will at some point, in the U.S. and elsewhere, be worth examining death rates across different areas, as it would be helpful to know if climate or other abiotic variables might impact this statistic. Coupling regularly updated data with automated online resources would also be particularly helpful in learning how this virus may spread.

## CONCLUSIONS

Species distribution models from SARS-CoV-2 population-scaled cases did not appear to be distinguishable from human population density for an early point in the pandemic for the U.S. Future studies looking at climate's impact on this virus should, wherever possible, take into account human population density in any analyses.

## ACKNOWLEDGEMENTS

We thank those who are treating the sick, providing essential services, preventing viral spread, quarantining alone, and helping others during this time of need; we thank the New York Times for making their data publically available; we thank the other researchers who have been providing a lively debate on best practices and how to make predictions at a point in time when our methods may matter most; and we thank Peter Galante and Rob DeSalle for reviewing a draft of this manuscript.

### Funding

The authors received no funding for this work.

### Competing Interests

The authors have no competing interests to declare.

### Author Contributions

- Robert Harbert conceived and designed the experiments, performed the experiments, analyzed the data, prepared figures and/or tables, authored or reviewed drafts of the paper, and approved the final draft.
- Seth W. Cunningham conceived and designed the experiments, performed the experiments, analyzed the data, authored or reviewed drafts of the paper, and approved the final draft.
- Michael Tessler conceived and designed the experiments, performed the experiments, analyzed the data, authored or reviewed drafts of the paper, and approved the final draft.

### Data Availability

Raw analysis code repository is available at GitHub: https://github.com/rsh249/cv19_enm.

### Supplemental Information

Supplemental information for this article can be found online at http://dx.doi.org/10.7717/peerj.10140#supplemental-information.

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
