# Peer review of "Spatial modeling could not differentiate early SARS-CoV-2 cases from the distribution of humans on the basis of climate in the United States"

_PeerJ, doi:10.7717/peerj.10140_

## Round 0.1 · original submission · Major Revisions

The topic of the proposed manuscript is interesting but there are currently weaknesses in the use and description of methods, as highlighted by the reviewers. I believe that the substantial comments they give would allow the authors to propose a much-improved version of their study, with appropriate application and discussion of the methods, and appropriate reference to the literature. The authors have also to improve the general presentation of the study, its consistency, and the quality of the writing.

I also support the idea that SDMs are inappropriate tools for SARS-CoV-2 coronavirus distribution. But since your conclusion is "climate may not play a central role in current US viral distribution and that human population density is likely a primary driver", I give you a chance to address reviewers' comments and also highlight the doubts/limitation about the modelling approach you used.

You might cite the below paper and expand your discussion accordingly.

https://www.nature.com/articles/s41559-020-1212-8

Reviewer 1 ·

Basic reporting

N/A

Experimental design

N/A

Validity of the findings

N/A

Additional comments

N/A

Reviewer 2 ·

Basic reporting

This paper seeks to determine whether climate data can contribute to predicting the range of SARS-CoV-2 in addition to human population distribution patterns. They found little evidence that climatic variables contribute much predictive power not explained by human population distribution. The paper is written with appropriate caution and the authors outline the caveats well.

However, there is much detail missing from the Methods, and the wording is so confusing that it's difficult to determine exactly what was done. This is despite the authors' own claims about the good level of methodological information they reported.

Further, some crucial elements were missing or lacking from either the Introduction or Discussion:
1) a clear acknowledgment that viral distribution patterns are also heavily driven by regional politics (i.e., country-specific responses to public health). Doesn't this also need to be accounted for (especially in the US)? High transmission rates, regardless of climate or human behavior, may be higher somewhere just because a particular gov't did not prepare sufficiently, or because they are reporting cases more accurately, etc.
2) a brief review of how SDMs have been used to model pathogen distributions, the difficulties they faced, how they overcame them, etc.

As a whole, although the topic may be of interest to those studying SARS-CoV-2 distributional patterns and the paper does call out some preprints claiming that the viral distribution can be properly explained by climate, key elements are missing that make for potential confusion, and these need to be addressed.

Experimental design

The main research question was well-defined, but the way key results were referred to was inconsistent (please use the same names for data products throughout), key methodological details were missing, and some confusion regarding model evaluation appeared to exist. I explain in the line-by-line comments, but selection of a model via AIC does not consider cross validation results, despite the authors' claims they used spatial cross validation to choose models. Table 1 is appreciated, but these same details must be referred to somewhere in the Methods text.

Validity of the findings

This paper does help to dispel the notion that viral distributions can be predicted solely with climate, but the analysis itself does not include enough human-specific variables to make a convincing model prediction. For example, some effect of regional politics is sure to play a role in infection rates, especially in the US, but this is glossed over in a single sentence in the Discussion. Nonetheless, the findings of this study will contribute overall to the understanding of the biogeography of SARS-CoV-2, and the authors are entirely transparent about data sources.

Additional comments

Line-by-line comments:

L65: Change to: "appear to explain much of the viral distribution".
L69-70: If SARS-CoV-2 is dependent on human hosts, what kinds of abiotic variables might shape its niche? Wouldn't most of these niche factors relate to the host's internal environment? How would knowledge of the "abiotic preferences" of the virus help contain its spread. Please explain more clearly.
L73: Remove "of".
L76: Maybe "foundation" instead of jumping-off point?
L86: lipid-enveloped
L112: Remove colon
L121: Isn't it "to at least some extent reflect the habitat patterns of their host"?
L136: self-isolation
L140: Needs updating
L164: Change to "interpolated climate data". Can you at least verify that using a more up-to-date interpolated climate dataset like CHELSA would not have affected results very much? More importantly, how good is Worldclim at predicting current weather? This pandemic just occurred. Wouldn't using PRISM have made more sense?
L166: Min and max of what? Avg monthly? Of warmest quarter, etc.?
L175: What is the purpose of this section? Please outline the logic behind your methods at the end of the intro.
L177: Probability of what? What exactly is the function of this kernel density grid? Why does it use climate variables? It looks from Fig. 1 that you took the kernel densities of cases and human pop, not climate (which would not make sense anyway). At this stage in the paper I am confused.
L180: population-scaled
L189: By "climate record", do you mean county record associated with climate? Also, what is a "raw SDM"?
L190-193: I don't follow this at all. What does it mean to "expand" a record? Why would any counties have fewer than one data point?
L195: It is not clear what the difference between these two is. Please explain more clearly. Also, where is the description of the "human population SDM"? Not even sure what that means, as SDMs are based on presence data, not abundance.
L195-199: It looks like you selected model settings for Maxent based on two datasets (but not clear what you did if the two were different), then used these settings to make models with other datasets. This makes little methodological sense, so I think I must be misunderstanding. Please clarify. Also, was maxnet used in your ENMeval run?
L202: Please explain briefly why this is preferred for models with regularization. From my understanding, it is best practice for the purposes of interpretation and to avoid extrapolation errors due to changes in covariance. Also, where do you discuss model evaluation?
L205: Please rephrase for correct grammar and fix spelling.
L209: Please explain the niche overlap and similarity tests. What do they do? How are they different?
L218: record(s)
L238: Please clarify "for people". As there are multiple variables considered, please reference them explicitly to avoid confusion.
L271-272: The model does not indicate that raw values should not be used as inputs. You infer that after analyzing the results.
L302-304: This seems like a relatively trivial exercise given you have the data in hand. Why didn't you do this and briefly compare it to your other results?
L324:324: If we are missing lots of data from more tropical areas, this would certainly affect global climate models. Why such uncertainty?
L325: What "best practices" are you referring to? The lead author on the paper criticized is the same one in the best practices paper you cite. Shouldn't this person, of all people, adhere to best practices? That seems like a valid criticism.
L328: A number of these best practices are missing from your Methods.
L351: Needs updating: https://www.who.int/news-room/commentaries/detail/bacille-calmette-gu%C3%A9rin-(bcg)-vaccination-and-covid-19

Table 1. Using AICc for model selection does not consider the results of cross validation at all. So regardless of the fact that you used spatial block cross validation with ENMeval, you did not consider the cross validation results unless you also considered test stats like testing AUC or omission rates, etc. Key details like which Maxent parameters you considered, etc., are missing from the Methods.

Fig 1. Nowhere in the text do you explain the relevance of wind speed, water vapor pressure, or solar radiation to SARS-CoV-2 distribution.

Reviewer 3 ·

Basic reporting

no comment

Experimental design

no comment

Validity of the findings

no comment

Additional comments

The authors use an SDM model to compare suitability for SARS-CoV-2 with human population suitability. The aim of the study is partly to critique earlier work using similar models which found a more conclusive effect of climate on SARS-CoV-2. I think the paper is well-written, well-argued and important. I recommend the study be published by PeerJ. I have only minor comments:

92 is this “scientists” or ecologists or some other sub-group? There are lot of scientists who do not use these methods so I think you need to be more specific.

139, remove “our” here as you haven’t introduced your model yet, suggest “While epidemiological models are highly uncertain, COVID-19 may peak…”

141 This sentence is confusing. It seems you are trying to state the purpose of the paper here? “Baseline” could mean a lot of things. I think the purpose of the paper is better stated in the abstract and some version of this should be re-stated here e.g. “Here we develop a SDM model using both climate and human population data to determine whether the effect of climate can be appropriately disentangled from other drivers…” – some better version of this.

155 I think the authors should clearly state that they are only take one value per county: total cumulative cases as of March 30th. Are call counties reporting? Would be nice to know the sample size.

169 5km buffer is very small, aren’t most counties much larger than this? You could have used a county shapefile to take averages over the climate data. Or by county center do you mean the population center? If you mean the county centroid, then that isn’t necessarily representative of where people live? In general, I don’t think this will matter too much, as climate data tends to be highly correlated over short distances. Perhaps mention this to justify using the small area.

176 (similar to line 155 comment) Are you using the total cases as of March 31ST 2020? State this clearly. Lots of the climate-regression studies use the whole time series of cases and compare variations in county-level climate with county-level cases over time. I think you need to make clear that you are using a purely cross-sectional approach.

Discussion: Probably worth citing this recent paper in Science suggesting even if climate can impact transmission, it will have minimal impact compared to population susceptibility. This may explain higher case numbers in Florida and other warmer locations.

Baker, Rachel E., et al. "Susceptible supply limits the role of climate in the early SARS-CoV-2 pandemic." Science (2020).

---

## Round 0.2 · Minor Revisions

Dear authors, thanks for taking our comments and providing us with the revised files. Your paper has been reviewed and the comments of the reviewer are included at the bottom of this letter. The reviewers and I have found your revision quite satisfactory and recommended publication, but also suggests some minor revisions to your manuscript. Therefore, I invite you to respond to the reviewer's comments and revise your manuscript.

Thank you

Reviewer 2 ·

Basic reporting

The paper is much easier to follow now, but there are still some sentences phrased awkwardly, grammatical mistakes, and unclear statements. I made suggestions in my line-by-line edits. Much better coverage of literature now.

Experimental design

This revision made the methods more clear, but lingering questions remain (see line-by-line edits). The authors need to use consistent language when referring to results, explain the methods in a bit more detail, better outline the methods in the introduction to prepare the reader for them, and in general take care to be consistent with the order and manner in which different results are reported (else it gets confusing).

Validity of the findings

The findings here are important for the field, as the authors demonstrated that climate-based SDMs are not appropriate to model the range of a novel pathogen spread by humans, and that human population density is a much better predictor. The authors just need to be more emphatic that this is the best way forward. The conclusions are clear and there is an appropriate amount of uncertainty outlined.

Additional comments

Good job on this latest revision -- I think the paper has improved considerably.

Line-by-line comments:

L29: "not in population equilibrium"
L32: "population-scaled" -- all instances need hyphens
L52: What is a "basic distribution"?
L70: Spelling
L97: Format
L102: Please finish the sentence.
L103: If the abbreviation SDM refers to "modeling", then SDMs becomes "species distribution modelings".
L106: Please cite something more recent and general, like 2011 Peterson et al. book.
L109: Grammar
L108-113: Not clear what the difference here is. A model of potential distribution is making correlations with climate data. Please be more clear about the distinction you are drawing.
L117: I would say "environmental variables correlated with its occurrence exist" because you can project to unsuitable conditions too.
L129: No capitalization mid-sentence.
L130: Why do you cite the standards paper here? Seems not relevant.
L142: "Still, the climate-based SDMs for SARS-CoV-2 presented recently..."
L143: Can we really use "habitat preferences" for modern humans? Can't you say "climatic preferences"? This is more accurate anyway because climate does not define habitat.
L146: "..., and may help to better ensure..."
L158-159: You do not mention here the niche overlap analyses. Please prepare the reader adequately for all the methods you undertake and explain why you're doing them.
L204: "were produced" -- please make sure to stay in past tense for the Methods section
L209: If "total cases per county" = "raw case data", say it like this to avoid confusion "Raw case data (i.e., total cases per county)"
L215: "we calculated non-parametric Spearman's rank correlation coefficients in R" (the code is not necessary)
L221-223: Rephrasing like this may be less confusing, and includes the expansion factor: "while occurrence data for population-scaled SDMs were generated the same way as for the raw SDM except that county climate records were multiplied by the total case count divided by the county population, such that no county had fewer than one record, then multiplied by an expansion factor." Please why you use the expansion factor.
L224: Use "total virus cases", not just "cases".
L227: Please cite Maxent here and introduce it as an SDM algorithm for presence-only data with complexity settings that can be tuned.
L230: After "feature classes", put "(constraints on model fit)" and after "regularization multipliers", put "(penalizes complexity)" for those who don't already know.
L231: Not "best", but "optimal", which emphasizes that you did not pick the absolute best but the best of the set explored.
L232: Rephrase to: "Optimal model parameters were chosen by maximizing the average test AUC calculated with cross validation using the spatial ‘block’ partitioning method, and minimizing AICc in the case of ties."
L234: "including those tested within ENMeval" -- not needed?
L236: Does "human density" here mean "human population"? Later in the Results you refer to "human population model" -- is that this human density SDM? Please use consistent terms.
L237: What does "clearly linked" mean? Can't you easily quantify their correlation?
L244-245: It is crucial to mention here which particular methodological decisions brought you to bronze, else readers cannot appreciate what is so commendable about your study.
L245: "inherently imperfect" is very vague. Can you be more specific as to why this data is imperfect for SDMs?
L260: "during the latest date in March" -> "for March 30"
L280: What are the results for the human pop model? They should be here.
L287: Not sure this interpretation is correct. I would interpret mean test AUC as a measure of average transferability and the variance as variation in transferability, as the AUC itself is measuring how well the trained model predicts the new data. Further, a difference of 0.0068 in variance is very small -- perhaps "slight difference"?
L290: Please report the results for the human pop model -- they seem to be missing. What were the optimal model settings? Also, I think they belong in the preceding paragraph where you discuss the pop scaled case model.
L311: "At least some human-focused data" is not really strong enough here when you just demonstrated that the observed infection patterns reflect human population density. Shouldn't this be the main kind of variable used?
L315-316: "inline" should be "in line"
L336: What does "statistically identical" mean? Aren't these different models with different complexity settings and different training data?
L338: Please quantify this, as the data exists.
L341: The models are different, and are certainly "statistically distinguishable" even though the niche similarity test was significant. Figure 3 shows the differences clearly. You might say they are "very similar".
L371: Which papers? Do you mean specifically those doing SDM analyses, or those doing SDM analyses on disease vectors? Please be more specific.
L392: Just say "individuals".
L398: Perhaps "while conducting an analysis on global data would have been ideal, we avoided this because..."
L435: What "heartbreaking statistic"? Might want to avoid using such emotional language.

Reviewer 3 ·

Basic reporting

no comment

Experimental design

no comment

Validity of the findings

no comment

Additional comments

I am happy with the authors' response to my questions and the changes made. I suggest the authors double-check spelling/grammar before submitting the final version e.g. Line 102 "once the endemic phase is reached" and Line 315 inline should be in line.

---

## Round 0.3 · accepted · Accept

I am delighted to accept your paper for publication in PeerJ.

Well done.